# Prediction of Conversion from Mild Cognitive Impairment to Alzheimer’s Disease Using Amyloid PET and Brain MR Imaging Data: A 48-Month Follow-Up Analysis of the Alzheimer’s Disease Neuroimaging Initiative Cohort

**DOI:** 10.3390/diagnostics13213375

**Published:** 2023-11-02

**Authors:** Do-Hoon Kim, Minyoung Oh, Jae Seung Kim

**Affiliations:** 1Department of Nuclear Medicine, Asan Medical Center, University of Ulsan College of Medicine, Seoul 05505, Republic of Korea; k8016851@eulji.ac.kr (D.-H.K.); my@amc.seoul.kr (M.O.); 2Department of Nuclear Medicine, Daejeon Eulji Medical Center, Eulji University School of Medicine, Daejeon 35233, Republic of Korea

**Keywords:** positron emission tomography, magnetic resonance imaging, Alzheimer’s disease, shape feature, Alzheimer’s disease neuroimaging initiative cohort

## Abstract

We developed a novel quantification method named “shape feature” by combining the features of amyloid positron emission tomography (PET) and brain magnetic resonance imaging (MRI) and evaluated its significance in predicting the conversion from mild cognitive impairment (MCI) to Alzheimer’s disease (AD) in the Alzheimer’s Disease Neuroimaging Initiative (ADNI) cohort. From the ADNI database, 334 patients with MCI were included. The brain amyloid smoothing score (AV45_BASS) and brain atrophy index (MR_BAI) were calculated using the surface area and volume of the region of interest in AV45 PET and MRI. During the 48-month follow-up period, 108 (32.3%) patients converted from MCI to AD. Age, Mini-Mental State Examination (MMSE), cognitive subscale of the Alzheimer’s Disease Assessment Scale (ADAS-cog), apolipoprotein E (APOE), standardized uptake value ratio (SUVR), AV45_BASS, MR_BAI, and shape feature were significantly different between converters and non-converters. Univariate analysis showed that age, MMSE, ADAS-cog, APOE, SUVR, AV45_BASS, MR_BAI, and shape feature were correlated with the conversion to AD. In multivariate analyses, high shape feature, SUVR, and ADAS-cog values were associated with an increased risk of conversion to AD. In patients with MCI in the ADNI cohort, our quantification method was the strongest prognostic factor for predicting their conversion to AD.

## 1. Introduction

Significant efforts are underway to identify and develop reliable biomarkers for Alzheimer’s disease (AD), to allow the targeting of those individuals who would most benefit from early treatment intervention [1], particularly those with mild cognitive impairment (MCI) [2,3]. However, because the etiology of MCI is heterogeneous, the rate of cognitive decline varies considerably across patients with MCI, and some do not convert to AD [4]. Thus, identifying patients with MCI who would benefit from treatment is essential [5]. It is widely assumed that imaging biomarkers for predicting conversion from MCI to AD are grouped into those based on beta-amyloid deposition, pathological tau, or neurodegeneration [6]. Amyloid positron emission tomography (PET) radiotracers, such as F-18 flutemetamol, F-18 florbetapir (AV-45), and F-18 florbetaben, have been developed for the assessment of beta-amyloid deposition in the brain [7,8,9]. Additionally, measures of brain atrophy on magnetic resonance imaging (MRI) have been used as biomarkers of neurodegeneration or neuronal injury [6,10,11].

As MCI progresses, brain atrophy increases and the shape of the brain changes on MRI. However, biomarkers of neurodegeneration cannot be used directly to represent the pathophysiological process of AD because they exhibit topographical overlap with non-AD pathologies [6,12]. Amyloid PET is correlated with the presence and density of amyloid deposition [8,9] and has been shown to be a predictor of future cognitive decline [13]. As amyloid deposition increases, the cortical uptake of amyloid radiotracers increases, and cortical gray–white matter differentiation is no longer possible on amyloid PET images. Furthermore, the cortical shape changes to a smoother form, and the differentiation of the gyral cerebrospinal fluid space on amyloid PET images is reduced by partial volume effects [14,15]. Studies have used each biomarker type to predict the conversion from MCI to AD; however, only a few studies have integrated amyloid and neurodegenerative markers. Obtaining integrated biomarkers that combine information from both beta-amyloid deposition on amyloid PET and brain atrophy on MRI would enhance the predictive power of these biomarkers. Therefore, developing quantification methods that combine information from various biomarkers is necessary. We hypothesized that the predictive power of biomarkers would be significantly enhanced if we used an integrated biomarker that combined information from beta-amyloid deposition on amyloid PET and brain atrophy on MRI.

In this study, we developed a novel shape feature quantification method for AV-45 PET and brain MRI and evaluated its prognostic significance in predicting the conversion from MCI to AD in patients from the Alzheimer’s Disease Neuroimaging Initiative (ADNI) cohort.

## 2. Materials and Methods

### 2.1. Patients

Data were obtained from patients recruited to the ADNI with available baseline data on AV-45 PET and MRI (http://ida.loni.usc.edu accessed on 6 September 2023). The ADNI was launched in 2003 as a public–private partnership, under the guidance of Principal Investigator Michael W. Weiner, MD, from VA Medical Center and the University of California in San Francisco, CA, USA. Patients were recruited from over 50 sites across the USA and Canada. The primary purpose of the ADNI was to test whether serial MRI, PET, other biological markers, and clinical and neuropsychological assessments could be combined to measure the progression of MCI and early AD (for up-to-date information, please refer to http://www.adni-info.org accessed on 6 September 2023). Written informed consent for cognitive testing and neuroimaging was obtained from all patients before their participation, and the procedures were approved by the Institutional Review Boards of all participating institutions.

For this study, 334 patients with MCI with baseline AV-45 PET data, baseline brain MRI data, and 4-year follow-up clinical evaluation data were selected. These patients were grouped into MCI converters or non-converters based on whether they had converted to AD within the 4-year follow-up period. The cognitive function of the patients was evaluated using the Clinical Dementia Rating Sum of Boxes (CDR-SB), Alzheimer’s Disease Assessment Scale–Cognitive Subscale (ADAS-cog), Functional Activities Questionnaire (FAQ), and Mini-Mental State Examination (MMSE).

### 2.2. AV-45 PET/CT and MRI

All PET and MRI data were retrieved from the ADNI database in the most advanced preprocessed stage. AV-45 PET images were acquired 50–70 min after the injection of F-18 AV-45 370 MBq (10 mCi); the images were then co-registered to each other, averaged across time frames, and standardized to the same voxel size (1.5 × 1.5 × 1.5 mm). The images were acquired at the 57 sites participating in the ADNI, and scanner-specific smoothing was applied. Because of the lack of scaling or warping processes, brain size and shape were not altered after preprocessing. For MRI, imaging was performed at 3T using T1-weighted imaging parameters. T1-weighted magnetization-prepared rapid gradient-echo sequences were used to correct image geometry distortion and image intensity nonuniformity, and a histogram peak sharpening algorithm was used. The postprocessed images used for the analysis in this study can be downloaded from the ADNI database.

### 2.3. Quantitative PET Image Analysis

The semiautomatic quantification of brain beta-amyloid deposition was performed using the brain amyloid smoothing score (BASS), which was calculated using the following formula:(1)$$BASS=\frac{Spherical\;surface\;area\;having\;the\;same\;volume\;as\;the\;VOI\;t50\%}{Surface\;area\;of\;VOI\;t50\%}$$
where VOI is the volume of interest and VOI t50% is the VOI segmented with a standardized uptake value (SUV) threshold of 50%.

The postprocessed images received from ADNI were converted to a file with the filename extension “nii”. The segmented brain was generated by combining segmented gray and white matter images using the SPM12 software package (https://www.fil.ion.ucl.ac.uk/spm/ accessed on 6 September 2023) running within MATLAB 2022a (MathWorks, Cambridge, UK). The algorithm was described in a unified segmentation paper [16]. A mask image was created using a gray matter image plus a white matter image with a threshold of 50%. Using the “regionprops3” function within MATLAB 2022a (MathWorks, Cambridge, UK), we calculated the surface area and volume of the mask image and selected the largest value. We calculated the volume and surface area of the AV45 image by considering the voxel size. The BASS value was derived using Formula (1).

The rationale for the definition of the BASS is as follows. Because a sphere is a three-dimensional object with the smallest surface area for a given fixed volume, a VOI with a smoother, more sphere-like surface would have a higher BASS value (Figure 1). When a receiver operating characteristic (ROC) value was calculated using BASS values and conversion using thresholds of 30%, 40%, 50%, 60%, and 70%, the area under the ROC curve (AUC) value was highest at the threshold of 50%, and this value was used (Figure A1).

To acquire the SUV ratio (SUVR) from the PET images, baseline structural MRIs were first co-registered with each participant’s AV-45 PET images. These images were then used to extract the mean weighted cortical retention uptake from the frontal, parietal, cingulate, and temporal regions. The SUVR was calculated using a composite reference region comprising the entire cerebellum, pons, and eroded subcortical white matter [17].

### 2.4. Quantitative MRI Analysis

A semiautomatic quantification of brain atrophy was performed by calculating the brain atrophic index (BAI) using the following formula:(2)$$BAI=\frac{Surface\;area\;of\;segmented\;brain}{Spherical\;surface\;area\;having\;the\;same\;volume\;as\;the\;segmented\;brain}$$

In MRIs, segmented gray matter and white matter images were also generated from postprocessed images received from the ADNI using the SPM12 software package (https://www.fil.ion.ucl.ac.uk/spm/ accessed on 6 September 2023) running within MATLAB 2022a (MathWorks, Cambridge, UK). A mask image was created using a gray matter image plus a white matter image. Using the “regionprops3” function within MATLAB 2022a (MathWorks, Cambridge, UK), we calculated the surface area and volume of the mask image and selected the largest value. We calculated the volume and surface area of the AV45 image by considering the voxel size. The BAI value was derived using Formula (2).

The rationale for the definition of the BAI is as follows. Because a sphere is a three-dimensional object with the smallest surface area for a given fixed volume, a VOI with a more irregular surface would have a higher BAI value (Figure 1).

### 2.5. Prediction of Cognitive Decline in Patients with MCI

A semiautomatic quantification of brain beta-amyloid deposition and brain atrophy was performed using the shape feature, which was calculated using the following formula:(3)Shape feature=BASS×BAI

When an ROC value was calculated using shape feature values and conversion using thresholds of 30%, 40%, 50%, 60%, and 70%, the AUC value was highest at the threshold of 50%, and this value was used (Figure A1). These longitudinal changes were calculated by comparing the measurements at 2-year follow-up visits with those acquired at baseline.

### 2.6. Statistical Analyses

Continuous data are expressed as means ± standard deviations, and categorical data are presented as frequencies. Continuous data analysis was performed using the independent sample *t*-test, and categorical data were analyzed using Pearson’s chi-square test. A comparison ROC curve was used to compare the parameters of the shape feature, BASS, BAI, and SUVR. Correlations between the shape feature and SUVR were assessed using Spearman’s rank correlation coefficient. ROC analysis was performed to identify the optimal conversion-predicting cutoff values for age, MMSE, ADAS-cog, apolipoprotein E4 (APOE4) levels, SUVR, AV45_BASS, MRI_BAI, and the shape feature. The univariate analysis of predictors of conversion was performed using the Kaplan–Meier method and log-rank test. A Cox proportional hazards model with a stepwise variable selection was used for multivariate analysis. Correlations between the shape feature and neuropsychological tests were assessed using Spearman’s rank correlation coefficient. MedCalc (Windows XP, version 12.3, Broekstraat, Mariakerke, Belgium) was used to perform all statistical analyses. *p*-values < 0.05 were used to denote statistical significance.

## 3. Results

### 3.1. Patient Characteristics

The mean age of the patients was 71.2 ± 7.1 years, and 108 (32.3%) of the 334 patients with MCI had converted to AD during the 48-month follow-up period. Conversion to AD was significantly associated with age, MMSE score, ADAS-cog score, and APOE4. No significant differences in sex and educational level were observed between converters and non-converters. The patient demographics are shown in Table 1.

### 3.2. Imaging Parameters

The mean SUVR, AV45_BASS, MR_BAI, and shape feature values were 1.22 ± 0.23, 0.40 ± 0.07, 3.98 ± 0.54, and 1.59 ± 0.33, respectively. Representative SUVR, AV45_BASS, MR_BAI, and shape feature examples are presented in Figure 1. SUVR, AV45_BASS, MR_BAI, and shape feature values were significantly higher in converters than in non-converters. The imaging parameters used in the 334 patients included in this study are summarized in Table 1.

### 3.3. ROC Curve Analysis

The usefulness of the SUVR, AV45_BASS, MR_BAI, and shape feature in the conversion to AD is presented by ROC curves (Figure 2 and Table 2). The ROC curves of the SUVR and shape feature were not significantly different (*p* = 0.994). Thus, the shape feature can be considered equivalent to SUVR. In contrast, a significant difference in the ROC curves was observed between SUVR and AV45_BASS (*p* = 0.001). SUVR was found to be a better parameter than AV45_BASS. Furthermore, a significant difference in the ROC curves was observed between SUVR and MR_BAI (*p* = 0.002), with SUVR being a better parameter than MR_BAI.

Figure 3 shows the correlation between SUVR and shape feature. SUVR and shape feature showed a significant positive correlation, with a correlation coefficient of 0.77 (r = 0.77; *p* < 0.0001).

### 3.4. Univariate and Multivariate Analyses for the Predictors of the Conversion from MCI to AD

The prognostic values of the patient characteristics and imaging parameters for predicting conversion from MCI to AD during follow-up are summarized in Table 3. The optimal cutoff values were determined using ROC curve analysis (Figure A2). A shape feature value of >1.54 (*p* < 0.001; hazard ratio [HR], 7.20; 95% confidence interval [CI], 4.77–10.88), an SUVR of >1.22 (*p* < 0.001; HR, 7.64; 95% CI, 5.05–11.56), an AV45_BASS of >0.41 (*p* < 0.001; HR, 5.22; 95% CI, 3.42–7.97), and an MR_BAI of >3.93 (*p* < 0.001; HR, 2.94; 95% CI, 1.96–4.42) were significant predictors of a shorter MCI duration, as were all other evaluated parameters.

The results of the multivariate Cox regression analysis adjusted for age, MMSE, APOE4, AV45_BASS, and MRI_BAI showed that the shape feature, SUVR, and ADAS-cog remained statistically significant predictors of a shorter MCI duration before conversion to AD. The HRs calculated for these three variables are shown in Table 4.

### 3.5. Correlation of the Shape Feature with Cognitive Outcomes

The shape feature calculated from the baseline PET and MR images of patients with MCI showed significant correlations with the longitudinal changes in cognitive measurements at 2 years (Figure 4). In particular, the shape feature was positively correlated with longitudinal changes in CDR-SB (r = 0.45; *p* < 0.001), ADAS-cog (r = 0.43; *p* < 0.001), and FAQ (r = 0.38; *p* < 0.001) and negatively correlated with longitudinal changes in MMSE scores (r = −0.36; *p* < 0.001).

## 4. Discussion

Using the data prospectively collected from the ADNI-2 cohort, we developed a novel shape feature quantification method that combines the baseline data of AV-45 PET and brain MRI and showed that this method could predict the conversion from MCI to AD. The shape feature was an important predictor of the conversion to AD because it had the highest HR in both the univariate and multivariate analyses. The shape feature could be used as a quantitative biomarker for predicting a longitudinal decline in cognitive measurements in patients with MCI and the conversion to AD. Our results demonstrate that the shape feature may be a feasible metric for identifying a clinically relevant semiquantitative biomarker.

Jack et al. proposed a framework for the in vivo staging of AD using two types of biomarkers—the measurement of beta-amyloid deposition and measurements of neurodegeneration [18]. According to the National Institute on Aging and Alzheimer’s Association Research Framework, AD is defined by its underlying pathological processes, which can be documented by postmortem examination or in vivo biomarkers [6]. This may help achieve a greater understanding of the mechanisms underlying heterogeneity and disease progression in AD [19]. However, beta-amyloid accumulation begins several decades before the appearance of the first cognitive symptoms, suggesting that the associations between these two types of biomarker abnormalities and the “time-dependent risk” of progression from MCI to AD vary considerably [20]. Therefore, determining a factor that can provide information on both amyloidosis and neurodegeneration is critical.

The prognostic value of amyloid PET using fluorinated tracers (visually [21], semiquantitatively [22], or both [23]) to determine cognitive decline and conversion to AD has been emphasized in patients with MCI, and its clinical impact on diagnostic confidence and drug treatment has recently been demonstrated [24]. Semiquantitative amyloid PET measurements commonly use the SUVR of cortical retention as a reference subcortical region [25]. However, the measurement of cortical retention in specific small cortical regions is limited by the partial volume effect that stems from the spatial resolution of PET scanners and post-smoothing images [15,16]. The disadvantages of using the SUVR are related to the variability of SUVR estimates, which can vary depending on the segmentation method used to define the target cortical and reference subcortical regions [26]. Because the BASS is not affected by a specific reference region, physicians can use this quantification method to generate consistent BASS values from a single AV-45 PET image. A limitation of the BASS is that even in cases with high degrees of amyloid deposition, the BASS may decrease if high-resolution PET images are used and brain atrophy has increased. However, in the shape feature method, such disadvantages of the BASS are compensated by measuring the structural changes using MRI.

Semiquantitative measures of atrophy on brain MRI commonly evaluate whole brain atrophy, hippocampal atrophy, or entorhinal cortex atrophy [27]. Guo et al. showed that the premorbid brain size was associated with protection against clinical deterioration in the face of AD-related brain atrophy in patients with MCI [28], thus supporting the theory that the brain reserve plays a compensatory role rather than a neuroprotective role [28]. Tabatabaei-Jafari et al. performed meta-regression analyses to investigate the impact of segmentation methodologies (manual vs. automated) on image-based atrophy measures and found that the manual segmentation of the hippocampus resulted in larger atrophy rate estimates than automatic segmentation using FreeSurfer [27]. Furthermore, Mulder et al. showed that lower atrophy rates can be achieved in investigations using automatic segmentation [29]. The BAI is a value obtained by measuring a large area and is thus expected to be less affected by segmentation variations than other methods; therefore, the BAI has excellent reproducibility with negligible interobserver and intraobserver variability.

The rationale for using the novel parameter AV45_BASS is that in positive scans, the full anatomical lobes are generally easier to visualize and the cortical margins are smoother [14]. Geometrically, the smooth shape of the beta-amyloid deposition leads to a low surface-area-to-volume ratio, which increases the sphericity of the surface. The rationale for using the novel parameter MR_BAI is that cortical margins show shrinkage in positive scans. Geometrically, this shrinkage of the brain cortex leads to a high surface area to volume ratio, which decreases the sphericity of the surface. Moreover, the MR resolution and limitations of the segmentation method may affect the use of the MR_BAI. Our novel shape feature is calculated by multiplying the AV45_BASS by the MR_BAI. One advantage of this feature over the SUVR is that it reveals an operator-independent characteristic of beta-amyloid deposition.

The correlation of the shape feature with neuropsychological tests may provide useful information on the timing and extent of beta-amyloid deposition, which are closely related to the clinical phenomenology of AD. Barthel et al. showed that the regional SUVR is well correlated with cognitive impairment measures, such as the MMSE, word-list memory, and word-list recall scores [30]. We noted significant correlations between the shape feature and longitudinal changes in neurological test results, including the CDR-SB (r = 0.45; *p* < 0.001), ADAS-cog (r = 0.43; *p* < 0.001), FAQ (r = 0.38; *p* < 0.001), and MMSE (r = −0.36; *p* < 0.001). These correlations raise the possibility of using AV-45 PET and brain MRI as markers of neuropsychological information.

Our study shows that the newly devised quantitative shape feature measurement may be used as a fusion biomarker for multimodal imaging. The demonstration of high amyloid deposition and high atrophy in the cortex at baseline on imaging biomarkers can predict longitudinal declines in cognitive scores [27,30]. However, determining a combined parameter that considers both amyloid deposition and atrophic changes is critical. The shape feature is a fusion biomarker directly obtained from both PET and MR images, which are also correlated with longitudinal cognitive measurements. This suggests that cognitive functions could rapidly deteriorate in patients with a high shape feature value at baseline. Therefore, the shape feature could act as a single parameter that reflects both PET and MRI patterns of future cognitive decline, particularly in terms of long-term outcomes. This correlation is an important observation with a potential impact on clinical trials for early treatment intervention in prodromal AD because the shape feature could help select patients likely to benefit from treatments.

## 5. Conclusions

We developed a new shape feature quantification method by combining the features of amyloid PET and brain MRI and showed that this shape feature was a strong imaging biomarker for predicting the conversion of patients with MCI to AD. Importantly, the shape feature values were significantly correlated with longitudinal changes in cognitive measurements in a sample of patients with MCI from the ADNI cohort. The shape feature measurement is expected to help identify patients with prodromal AD who may benefit from early intervention.

## Figures and Tables

**Figure 1 diagnostics-13-03375-f001:**
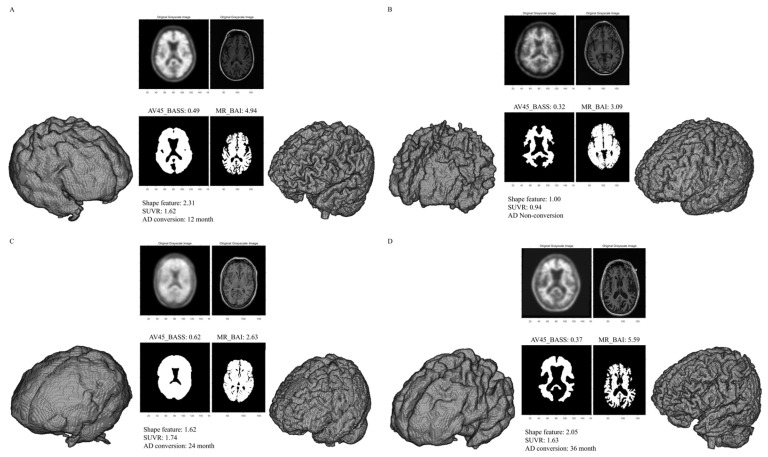
Representative segmentations used to measure the BASS and BAI. The volume of interest (VOI) from a patient with conversion to AD showed a smooth surface on AV-45 PET (high BASS score) and an irregular surface on brain MRI (high BAI score) (**A**). The VOI from a patient without conversion to AD showed a sharp surface on AV-45 PET (low BASS score) and a regular surface on brain MRI (low BAI score) (**B**). The VOI from a patient with conversion to AD showing a smooth surface on AV-45 PET (high BASS score) and a regular surface on brain MRI (low BAI score) (**C**). The VOI from a patient with conversion to AD showed a sharp surface on AV-45 PET (low BASS score) and an irregular surface on brain MRI (high BAI score) (**D**). Abbreviations: BASS, brain amyloid smoothing score; BAI, brain atrophic index; AV-45, F-18 florbetapir; SUVR, standardized uptake value ratio; AD, Alzheimer’s disease.

**Figure 2 diagnostics-13-03375-f002:**
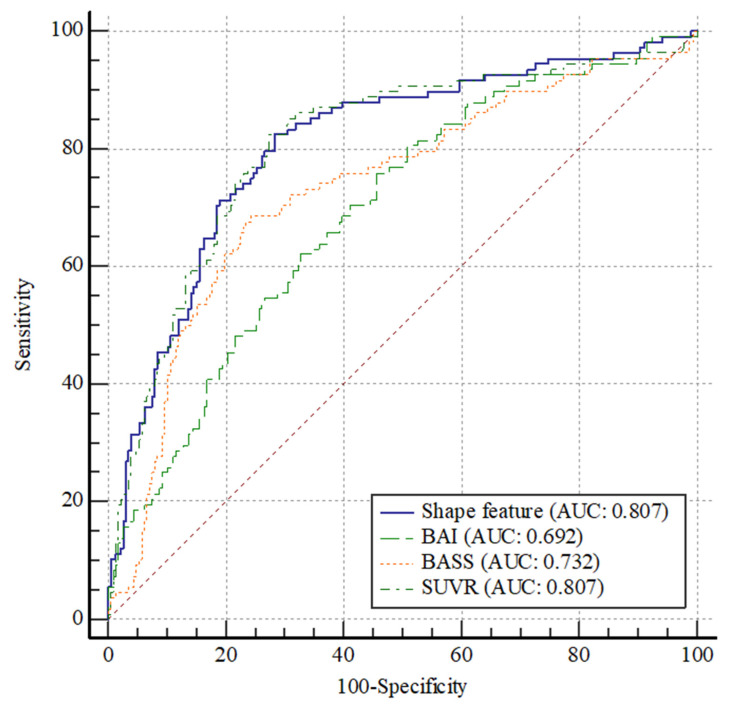
Comparison of ROC curves among the SUVR, AV45_BASS, MR_BAI, and shape feature. No significant difference in the ROC curves between SUVR and shape feature (*p* = 0. 994). Thus, they can be considered as comparable parameters. A significant difference in the ROC curves was observed between SUVR and AV45_BASS (*p* = 0.001); SUVR was a better parameter than AV45_BASS. A significant difference in the ROC curves was observed between SUVR and MR_BAI (*p* = 0.002); SUVR was a better parameter than MR_BAI. Abbreviations: ROC, receiver operating characteristic; AUC, area under the ROC curve; SUV, standardized uptake value; SUVR, standardized uptake value ratio; BASS, brain amyloid smoothing score; BAI, brain atrophic index.

**Figure 3 diagnostics-13-03375-f003:**
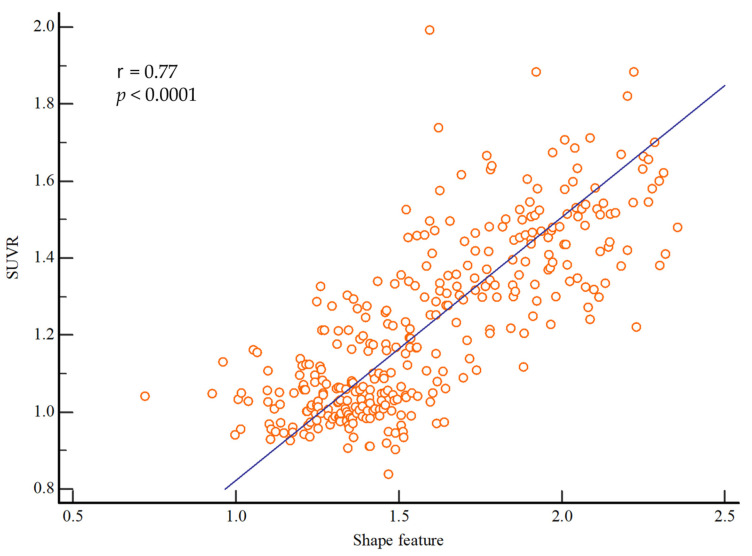
Correlations between the shape feature and SUVR values. There was a significant correlation between shape feature and SUVR (r = 0.77; *p* < 0.0001). Abbreviations: SUVR, standardized uptake value ratio.

**Figure 4 diagnostics-13-03375-f004:**
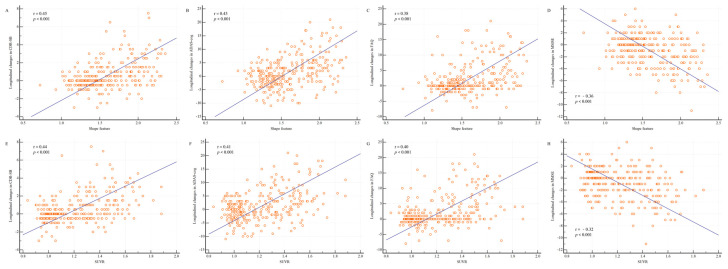
Shape feature and SUVR distributions according to longitudinal changes in cognitive measurements. Changes in the cognitive measurements between baseline and 2 years are shown. (**A**) Between the shape feature and Clinical Dementia Rating sum of boxes (CDR-SB); (**B**) between the shape feature and Alzheimer’s Disease Assessment Scale–Cognitive Subscale (ADAS-cog); (**C**) between the shape feature and Functional Activities Questionnaire (FAQ); (**D**) between the shape feature and Mini-Mental State Examination (MMSE); (**E**) between the SUVR and CDR-SB; (**F**) between the SUVR and ADAS-cog; (**G**) between the SUVR and FAQ; and (**H**) between the SUVR and MMSE. The shape feature was positively correlated with longitudinal changes in CDR-SB (r = 0.45; *p* < 0.001), ADAS-cog (r = 0.43; *p* < 0.001), and FAQ (r = 0.38; *p* < 0.001) and negatively correlated with longitudinal changes in MMSE scores (r = −0.36; *p* < 0.001). The SUVR was positively correlated with longitudinal changes in CDR-SB (r = 0.44; *p* < 0.001), ADAS-cog (r = 0.41; *p* < 0.001), and FAQ (r = 0.40; *p* < 0.001) and negatively correlated with longitudinal changes in MMSE scores (r = −0.32; *p* < 0.001). Abbreviations: SUVR, standardized uptake value ratio.

**Table 1 diagnostics-13-03375-t001:** Study population demographic characteristics.

Characteristics	Non-Converters	Converters	*p*-Value
Number	226	108	
Age (years)	70.5 ± 7.1	72.7 ± 6.8	0.01
Sex			0.90
Male	126 (55.8)	61 (56.5)	
Female	100 (44.2)	47 (43.5)	
Education (years)	16.4 ± 2.5	16.1 ± 2.6	0.26
MMSE	28.4 ± 1.6	27.3 ± 1.8	<0.001
ADAS-cog	12.2 ± 5.2	20.8 ± 6.8	<0.001
APOE4			<0.001
0	139 (61.5)	31 (28.7)	
1	73 (32.3)	58 (53.7)	
2	14 (6.2)	19 (17.6)	
SUVR	1.14 ± 0.18	1.39 ± 0.22	<0.001
AV-45_BASS	0.38 ± 0.06	0.44 ± 0.07	<0.001
MRI_BAI	3.86 ± 0.51	4.23 ± 0.52	<0.001
Shape feature	1.47 ± 0.28	1.83 ± 0.31	<0.001

Values are presented as means ± standard deviations or numbers (percentages). Abbreviations: MMSE, Mini–Mental State Examination; ADAS-cog, Alzheimer’s Disease Assessment Scale–Cognitive Subscale; APOE4, apolipoprotein E4; SUVR, standardized uptake value ratio; AV-45, F-18 florbetapir; BASS, brain amyloid smoothing score; BAI, brain amyloid index.

**Table 2 diagnostics-13-03375-t002:** Comparison of ROC curves.

Variable	AUC	95% CI	Comparison of ROC Curves between Each Variable and SUVR (*p*-Value)
SUVR	0.807	0.761–0.848	-
Shape feature	0.807	0.761–0.848	0.9936
AV45_BASS	0.732	0.681–0.779	0.0008
MR_BAI	0.692	0.640–0.742	0.0015

Abbreviations: ROC, receiver operating characteristic; AUC, area under the ROC curve; CI, confidence interval; SUVR, standardized uptake value ratio; BASS, brain amyloid smoothing score; BAI, brain atrophic index.

**Table 3 diagnostics-13-03375-t003:** Kaplan–Meier analysis of the conversion from MCI to AD and factors influencing the duration of MCI.

Variables	Hazard Ratio	95% CI	*p*-Value
Age (>69.3 vs. ≤69.3)	2.30	1.53–3.47	<0.001
MMSE (≤27 vs. >27)	2.41	1.24–4.68	0.009
ADAS-cog (>15 vs. ≤15)	6.89	4.54–10.46	<0.001
APOE4 (0 vs. 1 and 2)	3.12	2.08–4.69	<0.001
SUVR (>1.22 vs. ≤1.22)	7.64	5.05–11.56	<0.001
AV45_BASS (>0.41 vs. ≤0.41)	5.22	3.42–7.97	<0.001
MRI_BAI (>3.93 vs. ≤3.93)	2.94	1.96–4.42	<0.001
Shape feature (>1.54 vs. ≤1.54)	7.20	4.77–10.88	<0.001

Abbreviations: MCI, mild cognitive impairment; AD, Alzheimer’s disease; CI, confidence interval; MMSE, Mini-Mental State Examination; ADAS-cog, Alzheimer’s Disease Assessment Scale–Cognitive Subscale; SUVR, standardized uptake value ratio; AV-45, F-18 florbetapir; BASS, brain amyloid smoothing score; BASS, brain amyloid smoothing score; BAI, brain amyloid index.

**Table 4 diagnostics-13-03375-t004:** Multivariate Cox regression analysis of the conversion from MCI to AD and factors influencing the duration of MCI.

Variables	Hazard Ratio	95% CI	*p*-Value
Shape feature	5.70	3.40–9.56	<0.001
SUVR	6.06	3.66–10.04	<0.001
ADAS-cog	4.01	2.51–6.40	<0.001

Abbreviations: MCI, mild cognitive impairment; AD, Alzheimer’s disease; CI, confidence interval; SUVR, standardized uptake value ratio; ADAS-cog, Alzheimer’s Disease Assessment Scale–Cognitive Subscale.

## Data Availability

The ADNI dataset is available at: https://adni.loni.usc.edu accessed on 6 September 2023.

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
