# Peer review of "Prediction of Conversion from Mild Cognitive Impairment to Alzheimer’s Disease Using Amyloid PET and Brain MR Imaging Data: A 48-Month Follow-Up Analysis of the Alzheimer’s Disease Neuroimaging Initiative Cohort"

_diagnostics, 2023, doi:10.3390/diagnostics13213375_

Round 1
Reviewer 1 Report
Comments and Suggestions for Authors
Dr. Do-Hoon Kim and colleagues submitted to propose a “shape feature quantification method” for prediction of conversion from MCI to AD. This provides interesting findings, however there are several minor comments on this manuscript as follows.
1) What did a bright idea flash into your mind to combine information from beta-amyloid deposition on amyloid PET and brain atrophy on MRI for the better prediction of conversion from MCI to AD?
2) Table 1 reveals that the cognitive parameters, APOE4 level, SUVR of AV-45, AV-45_BASS, and MRI-BAI can significantly predict the conversion from MCI to AD. If you want to emphasize the more capability of the shape feature quantification method than these parameters, the correlative analyses should be conducted between each parameter and the longitudinal changes in the cognitive measurements as shown in Figure 3.
3) In Table 2, the authors compare between more and less of each threshold of the parameters to determine Hazard ratio, 95% CI, and P-value. It should be needed how to decide each threshold.
4) In Figure 3, the unit of Y-axis should be shown as, for example, “Longitudinal changes in CDR-SB” and so on. Otherwise, potential readers may misunderstand them as the parameters assessed at MCI stage.
5) In this proposed shape feature quantification method, authors combine AV-45_BASS, and MRI-BAI to calculate it. In 4. Discussion, at page 7, line 230, authors describe that “structural abnormalities are generally only visible on anatomical MRI after disease progression”, and also at page 8, line 257, “these protective effects of the morphologic brain reserve seem to be limited to early clinical AD”. These facts should cause the question how effective to apply MRI-BAI at MCI stage to predict the conversion from MCI to AD.
Reviewer 2 Report
Comments and Suggestions for Authors
The authors evaluated a quantitative method that combines measures of amyloid-PET and structural MRI in the prediction from MCI to AD in the ADNI database. The idea is interesting, and the paper was written well. However, the quantitative method that was developed and used by the authors was not explained enough.
1- The authors used a Brain Amyloid Smoothing Score (BASS) to evaluate brain amyloid deposition. The formula is the proportion of spherical surface area having the same volume as the VOI t50% to surface area of VOI t50%. It is not clear why the authors chose t50% cut-off while developing this formula. They mentioned that higher BASS represents more sphere surfaces but their example figures are in 2-D dimension and it's hard to imagine what they're trying to say. The authors should explain the rationale of this method better and compare it with the usual SUVR measures. They mentioned the disadvantage of this method in the discussion section as it might be affected by the resolution of PET images and brain atrophy. the reviewer is not convinced that it could be overcome by multiplying it with the Brain atrophy index (BAI). The reviewer also wonders whether the partial volume effect has been done during SUVR measures and whether the smoothing process was the same for all subjects. The reviewer has similar concerns about the BAI index (explain the rationale better, compare it with a usual measure of atrophy [cortical thickness, grey matter volume). Overall, the reviewer suggests adding a supplementary file to explain the methods used in detail with additional figures and comparison tables.
2- The authors should move the sentence about the results ('the shape feature was correlated.....") from the Section 2.5 to the results section where appropriate.
Round 2
Reviewer 2 Report
Comments and Suggestions for Authors
To the authors,
The authors provided adequate revisions based on the previous suggestions. The only comment is that the authors should adjust the resolution of Figure 4 as it's hard to read the titles and numbers.